# Developing a Biomimetic 3D Neointimal Layer as a Prothrombotic Substrate for a Humanized In Vitro Model of Atherothrombosis

**DOI:** 10.3390/biomimetics9060372

**Published:** 2024-06-20

**Authors:** Jassim Echrish, Madalina-Ioana Pasca, David Cabrera, Ying Yang, Alan G. S. Harper

**Affiliations:** 1School of Medicine, Keele University, Keele ST5 5BG, UK; 2School of Pharmacy and Bioengineering, Keele University, Keele ST5 5BG, UK; d.c.cabrera@keele.ac.uk (D.C.); y.yang@keele.ac.uk (Y.Y.)

**Keywords:** atherosclerosis, biomimetic hydrogel, animal use alternatives, vascular tissue engineering, thrombosis, platelets, coagulation

## Abstract

Acute cardiovascular events result from clots caused by the rupture and erosion of atherosclerotic plaques. This paper aimed to produce a functional biomimetic hydrogel of the neointimal layer of the atherosclerotic plaque that can support thrombogenesis upon exposure to human blood. A biomimetic hydrogel of the neointima was produced by culturing THP-1-derived foam cells within 3D collagen hydrogels in the presence or absence of atorvastatin. Prothrombin time and platelet aggregation onset were measured after exposure of the neointimal models to platelet-poor plasma and washed platelet suspensions prepared from blood of healthy, medication-free volunteers. Activity of the extrinsic coagulation pathway was measured using the fluorogenic substrate SN-17. Foam cell formation was observed following preincubation of the neointimal biomimetic hydrogels with oxidized LDL, and this was inhibited by pretreatment with atorvastatin. The neointimal biomimetic hydrogel was able to trigger platelet aggregation and blood coagulation upon exposure to human blood products. Atorvastatin pretreatment of the neointimal biomimetic layer significantly reduced its pro-aggregatory and pro-coagulant properties. In the future, this 3D neointimal biomimetic hydrogel can be incorporated as an additional layer within our current thrombus-on-a-chip model to permit the study of atherosclerosis development and the screening of anti-thrombotic drugs as an alternative to current animal models.

## 1. Introduction

Acute cardiovascular events are a leading cause of death and disability globally [1]. These thrombotic disorders occur following the rupture or erosion of atherosclerotic plaques that expose the bloodstream to highly thrombogenic biomaterial that has accumulated within the neointimal lining of the artery. The ensuing activation of the hemostatic system generates occlusive thrombi that block the downstream coronary or cerebral arteries that lead to heart attacks and strokes, respectively. Being able to study the events underlying human plaque disruption and the resulting atherothrombosis is key to our ability to develop treatments to prevent plaque disruption or to reduce the thrombotic response to limit ischemic damage to downstream tissues.

The natural history of atherosclerosis in humans is a complex, slow-developing, and unpredictable process that cannot be readily studied by non-invasive imaging techniques [2]. This begins with the accumulation of foam cells in the subendothelial space of an inflamed segment of the endothelium, creating a new neointimal layer that lies between the intimal and medial layers of the artery [3]. The recruitment of foam cells produces a fatty streak. As these cells undergo apoptosis in the crowded neointimal lining, this creates a necrotic core that acts as a proinflammatory niche that triggers smooth muscle cells migration to the neointima to form a fibrous cap that mechanically strengthens the plaque. The ongoing inflammatory conditions trigger smooth muscle cell apoptosis that leads to cap thinning and eventual rupture of the plaque, triggering thrombus formation. The resultant clot can become lodged in both coronary and cerebral circulation, resulting in myocardial infarction or stroke.

Our knowledge about the pathology and pathogenesis of atherosclerosis has principally been obtained from in vivo animal studies in mice, rabbits, pigs, and non-human primates [4]. However, there are significant differences in the lipid metabolism, vessel morphology, blood flow patterns, and immune responses of humans and these model species [5,6]. This has led to the development of many different animal models, each representing different aspects of human pathology [5,7]. Critically, many of the animal models used do not produce atherosclerotic plaques that progress to advanced-stage lesions that spontaneously erode or rupture—thus failing to replicate the clinical event that triggers myocardial infarction and strokes [5,6]. This significantly limits the ability of these models to be used in pre-clinical testing of anti-thrombotic drugs and are instead principally used for basic understanding of how plaques form, develop, and rupture. Those animal models that do demonstrate plaque rupture are unpredictable—with the most effective mouse models demonstrating a 50–75% rupture rate, occurring irregularly over prolonged study periods that last many months [8,9]. This makes studies time-consuming, costly, and reliant on the tracking of severe symptoms of plaque rupture that cause significant suffering to the animal. An alternative approach has been to use intravital microscopy in ApoE^-/-^ mice to monitor thrombotic response after the plaque was artificially disrupted using ultrasound, FeCl_3_, photochemical injury, ligation, or needle injury [9]. However, previous work has demonstrated that various responses are observed depending on the method of injury used in these atherothrombosis models [9,10]; therefore, it is uncertain whether these results are representative of the clotting responses experienced in patients. These limitations have prevented the adoption of these animal models in preclinical testing of anti-thrombotic drugs. 

The most direct method to replicate the atherothrombotic response elicited by human plaques would be to test this from plaque material removed during a variety of cardiovascular surgeries (e.g., carotid endarterectomy or repair of an abdominal aortic aneurysm), and utilizing this in an in vitro flow model. However, the availability of surgical material and the difficulties in obtaining a homogenous, reproducible coating on different slides has prevented this technique from being widely adopted [11,12,13]. Additionally, the homogenization of the plaque material may disrupt the altered collagen structures that have previously been shown to be responsible for the thrombotic response observed on these tissues, altering the response seen [11,12,14]. This has resulted in groups attempting to artificially recreate the development of atherosclerotic plaques within tissue-engineered human arteries [15,16,17,18,19]. However, these approaches have so far failed to reproduce anything beyond the build-up of foam cells within the wall of the artificial arteries—which represents an early stage, clinically silent stage of atherosclerosis (Type I/II of the American Heart Association (AHA) Classification [20]). This highlights the difficulty of replicating a slow, decades-long developmental process in tissue-engineered human arteries. It is uncertain whether this strategy can produce a clinically relevant, advanced-stage atherosclerotic lesion. 

An in vitro model of a human atherosclerotic plaque demonstrating properties of advanced-stage lesions was produced using hanging drop cultures by coating a microdroplet of myeloid cells with myofibroblast cells [21]. Whilst this study managed to effectively model the presence of the fibrous cap seen in more advanced atherosclerotic lesions, it did this in isolation, without other cells and layers found in the arterial wall, making it difficult to use as a model of atherothrombosis. However, this study demonstrated that culturing replicas of atherosclerotic plaques, in isolation from other components of the vessel wall, provide a greater ability to engineer aspects of advanced-stage lesions into the construct. Therefore, we considered that culturing a biomimetic model of the neointima that could be subsequently introduced between the intimal and medial layers of a tissue-engineered arterial construct may provide the optimal strategy for developing tissue engineering arteries containing advanced-stage atherosclerotic lesions.

Previously, we have developed a humanized in vitro arterial thrombosis model [22,23,24]. This model was produced using tissue-engineering techniques to produce a 3D human arterial construct. The intimal layer is created by culturing human umbilical vein endothelial cells (HUVECs) on top of fibronectin-coated aligned polylactic acid (PLA) nanofibers. Simultaneously, a medial layer construct was grown by culturing human coronary artery smooth muscle cells (HCASMCs) within a 3D type I collagen hydrogel. When complete, the intima and medial layer constructs can then be assembled to create the complete tissue-engineered human arterial construct (TEAC) that has both the anti- and pro-thrombotic properties of the intimal and medial layers of the native artery [22,23,24]. This layer-by-layer fabrication method used to produce the TEAC allows for each layer of the construct to be independently cultured prior to being brought together to produce the complete structure. This allows us to independently modify the culturing conditions of each layer prior to co-culture. This strategy also allows us to introduce a neointimal layer within this model system to produce an in vitro model of atherosclerosis. 

In this paper, we describe the production of a 3D neointimal biomimetic hydrogel of an early stage atherosclerotic plaque and demonstrate that, in isolation, it possesses the pro-thrombotic properties required to be effective as a substrate for the study of atherothrombosis. In the future, this neointimal biomimetic hydrogel will be incorporated as an additional layer into our current TEAC to produce a replica of a human atherosclerotic artery. This could then be used to produce a thrombosis-on-a-chip model as an alternative to current animal models.

## 2. Materials and Methods

### 2.1. Materials

Proliferating THP-1 cells (a human monocytic cell line derived from a leukemic patient) were obtained from the European Collection of Authenticated Cell Cultures (Porton Down, UK). Phorbol 12–myristate 13-acetate (PMA), paraformaldehyde and copper sulfate pentahydrate, Nile-Red, Oil Red O powder and Lipopolysaccharide (LPS), aspirin, apyrase, and luciferin–luciferase were purchased from Sigma-Aldrich (Gillingham, UK). RPMI 1640 with L-glutamine, fetal bovine serum (FBS), phosphate-buffered saline (PBS), Penicillin/Streptomycin (AA), and Whatman filter paper were purchased from Scientific Laboratory Supplies (Nottingham, UK). α-MEM medium powder, ethylenediaminetetraacetic acid (EDTA), Slide-lyzer dialysis cassettes (gamma-irradiated, 10K MWCO), and Corning low and high concentration rat tail collagen were purchased from Fisher Scientific (Loughborough, UK). Monoclonal recombinant human full-length CD80 protein antibody was purchased from Novus Biologicals (Abingdon, UK). The FITC-labelled goat anti-mouse secondary antibody, Human IL-6 and TNF-α ELISA Kits and oxLDL assay kit were purchased from Abcam (Cambridge, UK). Human IFN-γ was purchased from Peprotech (London, UK). Sterile low-density lipoprotein was purchased from 2B Scientific (Upper Heyford, UK). Live/Dead Staining Kit was obtained from PromoCell GmbH (Heidelberg, Germany). SN-17a was purchased from Cambridge Bioscience (Cambridge, UK). Human FVII was purchased from Enzyme Research Laboratories (Swansea, UK). Atorvastatin Calcium Trihydrate was obtained from Active Pharma Supplies Ltd. (Leyland, UK). 

### 2.2. Preparation and Validation of Oxidized Low-Density Lipoproteins

A commercially available preparation of sterile low-density lipoprotein (LDL) isolated from human plasma was diluted to 5 mg/mL using sterile PBS. Part of the stock was aliquoted and stored at −80 °C before its use in experiments as a non-oxidized LDL preparation. In the remaining stock, LDL oxidation was performed as previously described [25]. A total of 10 μM copper sulphate was added to the LDL solution and incubated for 24 h at 37 °C to allow for its oxidation. Excess copper sulphate was then removed from the sample solution through dialyzing the sample against a PBS buffer containing 100 μM EDTA for 24 h at room temperature under sterile conditions. Dialysis was performed using a Slide-a-Lyzer dialysis cassette (gamma-irradiated 10K MWCO; Fisher Scientific) according to the manufacturer’s instructions. Aliquots of this sample were then preserved at −80 °C before its use in experiments. The oxidized low-density lipoprotein samples were characterized through a comparison of the absorbance of the oxidized and native low-density lipoproteins at 300 nm, as well as through an oxidized LDL (oxLDL) ELISA assay kit from Abcam according to the manufacturer’s instructions. For the absorbance readings, oxidized and native LDL samples were loaded into a 96-well plate and their absorbance was read on a BioTek synergy 2 microplate reader (Agilent Technologies UK, Stockport, UK).

### 2.3. Preparation of the 3D Human Neointimal Cell Culture Construct

#### 2.3.1. To Trigger M0 Differentiation

THP-1 cells were mixed at a cell density of 3.6 × 10^6^ cells/mL into a neutralized solution of 3 mg/mL type I collagen. A total of 0.2 mL of the THP-1/collagen mixture was then loaded under sterile conditions onto a 1 × 1 cm (4 × 10^5^ cells per collagen hydrogel), sterile filter paper frame prepared on a PTFE base inside a Petri dish. The collagen–cell mixture was left to set for 30 min at 37 °C, 5% CO_2_. Then, the hydrogel was transferred to 6-well plates and covered with RPMI medium containing 10% [*v*/*v*] FBS, penicillin, streptomycin, and 50 nM PMA, and then incubated for 48 h at 37 °C, 5% CO_2_ to trigger differentiation of the cells into M0 macrophages.

#### 2.3.2. To Trigger M1 Differentiation

The M0-containing hydrogels were then placed into PMA-free, supplemented RPMI media (containing 10% [*v*/*v*] FBS, penicillin and streptomycin), and incubated for 24 h. The RPMI media were then changed to fresh supplemented RPMI media containing 100 ng/mL LPS and 20 ng/mL IFN-ɣ. The gels were then incubated at 37 °C, in a 5% CO_2_ incubator for 3 days.

#### 2.3.3. To Generate THP-1-Derived Foam Cells

The media were exchanged with fresh, supplemented RPMI media that contained either 50 μg/mL oxLDL, 50 μg/mL LDL, or an equivalent volume of PBS. The different samples of the collagen hydrogels were incubated at 37 °C, 5% CO_2_ for a further 3 days.

#### 2.3.4. Plastic Compression of 3D Neointimal Biomimetic Hydrogels

The collagen hydrogel was prepared as described previously. Two 1 × 1 cm filter paper frames were placed inside either 2 × 1 × 1 cm metallic or plastic molds (both are functionally equivalent). Two milliliters of hydrogel were put in each mold, and then placed inside 9.6 cm^2^ Petri dish. They were incubated in a humidified incubator at 37 °C, 5% CO_2_ overnight. The next day, the mold was flipped, and the gel extracted onto a nylon sheet with tissue paper underneath. A 50 g metal weight was applied atop the hydrogel for 5 min. The compressed collagen hydrogels were separated into two 1 × 1 cm frames [26]. Each compressed collagen hydrogel has 5 × 10^5^ cells. Thereafter, the compressed collagen hydrogels were transferred to 24-well plates containing completed RPMI media. The culture media were exchanged for PMA-free RPMI media, and the hydrogels were incubated for 24 h at 37 °C, 5% CO_2_. RPMI media were then exchanged again for fresh RPMI media containing 100 ng/mL LPS and 20 ng/mL IFN-ɣ was added, and then incubated for a further 3 days in an incubator at 37 °C, 5% CO_2_. The media were changed for fresh RPMI media containing either 50 μg/mL oxLDL or non- oxidized LDL and placed in the incubator for a further 3 days prior to experimentation. Where indicated, 20 μg/mL of atorvastatin, or an equivalent volume of its vehicle, methanol, was added alongside the oxLDL.

### 2.4. Assessment of Differentiation of THP-1 Cells within 3D Collagen Hydrogels

#### 2.4.1. Assessing Cell Viability of 3D Culture of THP-1-Derived Cells

After culture, the collagen hydrogels were transferred to 24-well plates and the collagen hydrogels were stained using the Live-Dead Staining Kit II (Promocell) according to the manufacturer’s instruction. Cell viability was assessed for fluorescent staining within the collagen hydrogels using a Fluoview FV 1200 laser scanning confocal microscope (Olympus, Southend-on-Sea, UK) using a 10× objective and with excitation at 473 nm (Calcein staining of live cells) or 543 nm (EthDIII staining of dead cells) and emission at 490–520 or 590–620 nm, respectively.

#### 2.4.2. Immunofluorescent Staining of the M1-Containing Hydrogels

Collagen hydrogels containing either M0 or M1 macrophages, respectively, were prepared as stated above. The hydrogels were then fixed with a 10% [*w*/*v*] formalin solution overnight at 4 °C. The fixatives were then removed, and the collagen hydrogels were blocked by the addition of HBS supplemented with 1% [*w*/*v*] bovine serum albumin (BSA) for 1 h at room temperature. The blocking solution was removed, and the collagen hydrogels were resuspended in a labelling solution made up of the blocking solution containing 0.2 mg/mL monoclonal human CD80 primary antibody. The collagen hydrogels were incubated with this staining solution at 4 °C overnight. The staining solution was removed, and the collagen hydrogels were washed three times with the blocking solution for five minutes at room temperature. After the third wash, the collagen hydrogels were stained by the addition of blocking solution containing 1:5000 FITC-conjugated anti-mouse IgG secondary antibodies for 1 h at room temperature. The collagen hydrogels were then washed 3 times with HBS containing 1% [*w*/*v*] BSA for five minutes at room temperature. The collagen hydrogels were then analyzed by confocal microscopy using the Fluoview FV 1200 laser scanning confocal microscope (Olympus, UK) using 473 nm excitation wavelength and emission at 490–520 nm. Fluorescent intensity was assessed using ImageJ.

#### 2.4.3. Fluorescent Imaging of THP-1 Derived Foam Cells in the 3D Neointimal Biomimetic Hydrogels

After culture, the tissue-engineered neointimal layer was fixed with a 10% [*w*/*v*] formalin solution overnight at 4 °C. The fixative was removed, and the collagen hydrogels were washed with PBS for 1 h at room temperature. Afterwards, the PBS was removed, and the collagen hydrogels were resuspended in 400 μL of PBS containing 100 μg/mL Nile Red. After incubation for 20 min at room temperature, the staining solution was removed, and the collagen hydrogels were washed with PBS. The samples were then imaged with a Fluoview FV 1200 laser scanning confocal microscope (Olympus, UK) using 473 nm excitation wavelength and emission at 490–520 nm.

#### 2.4.4. Measuring oxLDL Uptake into 3D Neointimal Cell Cultures

On the day of use, 0.5% [*w*/*v*] Oil Red O solution was dissolved in isopropanol and filtered. Each hydrogel was incubated with 500 μL of the Oil Red O solution for 30 min at room temperature on an orbital shaker (100 rpm). After this, the gels were washed three times with PBS. After this, the hydrogels were destained by addition of 500 μL isopropanol to each gel and rocked on an orbital shaker for 30 min. From this solution, 100 μL was placed into a 96-well plate and absorbance was read at 570 nm on a BioTek synergy 2 microplate reader to quantify the amount of neutral lipids present in the sample.

### 2.5. Assessment of the Hemostatic Effects of the 3D Neointimal Biomimetic Hydrogel

#### 2.5.1. Blood Donations

This study was approved by the Keele University Research Ethics Committee (Ref: MH-200158) and conducted in accordance with the Declaration of Helsinki. Blood was donated by healthy volunteers who had given written informed consent prior.

#### 2.5.2. Preparation of Human Platelet-Rich Plasma (PRP) and Platelet-Poor Plasma (PPP)

Whole blood was mixed in a 9:1 ratio with 3.8% [*w*/*v*] sodium citrate solution. The blood was centrifuged at 700× *g* for 8 min to separate blood into its constituents. Platelet-rich plasma (PRP) was aspirated and used in experiments. For preparation of PPP, 100 μM aspirin, and 0.1 U/mL apyrase was added, and the PRP was then centrifuged at 350× *g* for 20 min. The PPP was aspirated from the platelet pellet. The coagulation experiments were conducted using the supernatant of the PPP.

#### 2.5.3. Preparation of Washed Human Platelet Suspensions

Whole blood was mixed in a 5:1 ratio with acid citrate dextrose anticoagulant (85 mM sodium citrate, 78 mM citric acid, and 111 mM d-glucose). The PRP was obtained by centrifugation for 700× *g* for 8 min. After extraction, this was treated with 100 μM aspirin and 0.1 U/mL apyrase. Platelets were then collected by centrifugation at 350× *g* for 20 min. Washed platelets were re-suspended into supplemented HEPES –Buffered Saline (HBS) to a final cell density of 2 × 10^8^ cells/mL. To make supplemented HBS, HBS (pH 7.4, 145 mM NaCl, 10 mM HEPES, 10 mM d-glucose, 5 mM KCl, 1 mM MgSO_4_) was supplemented on the day of the experiments with 1 mg/mL BSA, 10 mM glucose, 0.1 U/mL apyrase, and 200 μM CaCl_2_. Immediately prior to all experiments, the CaCl_2_ concentration of the washed human platelet suspension was increased to 1 mM.

#### 2.5.4. Prothrombin Time Measurement

Citrated PPP (0.5 mL) was aliquoted into cuvettes and recalcified by addition of 20 mM CaCl_2_. The samples were then incubated for 10 min prior to the experiment in a water bath held at 37 °C. The tissue-engineered neointimal constructs were washed with PBS, and then inserted into sodium acetate frames to allow the surface of the neointimal constructs to be placed in contact with PPP (Figure 1). Prothrombin time was measured as the time taken between construct contact with PPP and the observation of the formation of fibrin crystals within the PPP sample.

#### 2.5.5. Extrinsic Pathway Clotting Factor Assay

Tissue factor activity of the tissue-engineered neointimal constructs was assessed by measuring tissue factor-dependent activation of FVII. THP-1 cells were mixed at a cell density of 3 × 10^6^ cells/mL into a neutralized solution of 3 mg/mL type I collagen, and then 200 μL of this sample was then set into sterile 48-well plates. These cells were then cultured in RPMI media over 10 days to foam cells, as detailed above through sequential addition of PMA, IFN-γ and LPS, and oxLDL or native LDL. On day 10 of the culture, the culture media were removed, and samples were monitored following the addition of 100 μL of a PBS solution containing 1 mM CaCl_2_, 100 nM FVII, and 10 μM SN-17a. Cleavage of SN-17a was then measured fluorometrically using a BioTek synergy 2 microplate reader using excitation wavelengths of 340–380 nm, and emission wavelengths of 518–538 nm. Readings were taken every 10 s for 10 min at 37 °C.

#### 2.5.6. Light Transmission Aggregometry

Platelet aggregation was measured using a ChronoLog light transmission aggregometer. The baseline for 0% aggregation was set at the start of experiment against a tube filled with 450 μL of unused washed human platelet suspension, and for 100% aggregation it was set using a tube filled with 450 μL of supplemented HBS. The neointimal constructs were placed into sodium acetate frames and placed into contact with 0.8 mL washed human platelet suspension. These were incubated for 10 min at 37 °C under constant magnetic stirring (Figure 1). Following this, the construct was removed, and in some experiments it was fixed in 4% paraformaldehyde. Then, 450 μL of the platelet suspension was transferred into glass aggregometry tubes containing magnetic stir bars. Stirring in the aggregometer was set to 1200 rpm and the temperature of the aggregometry tube holder was maintained at 37 °C. Changes in light transmission were recorded during constant stirring of the samples at 37 °C for up to 16 min.

#### 2.5.7. ATP Secretion Assay

A luciferin–luciferase assay was used to assess secretion of ATP from platelet-dense granules. After incubation with the neointimal construct for 10 min at 37 °C, the construct was removed, and samples of the washed platelet suspension were extracted for aggregometry analysis as stated in Section 2.5.6. The remaining washed platelet suspension was incubated without the construct for a further 10 min at 37 °C. A total of 80 μL of the washed platelet suspension was then placed into a 96-well plate and mixed with 20 μL luciferin–luciferase assay mixture (Sigma Aldrich, Gillingham, UK) immediately prior to reading. A BioTek Synergy 2 microplate reader was then used to measure ATP-induced luminescence from the sample over 1 min.

### 2.6. Statistical Analysis

All values are expressed as the mean ± SEM with the number of observations (n) indicated. Analysis of statistical significance was performed using either a two-tailed Student’s *t*-test or a one-way ANOVA followed by a post hoc Tukey test. A *p* value of <0.05 was considered statistically significant.

## 3. Results

### 3.1. Development of a 3D Neointimal Biomimetic Hydrogel

To produce an in vitro three-dimensional atherosclerotic plaque layer to be incorporated inside our tissue-engineered human arterial constructs, experiments first sought to establish a reliable methodology for culture of macrophage-derived foam cells within a 3D collagen hydrogel. Two distinct strategies were considered for developing the 3D neointimal biomimetic hydrogel. The first was to establish 2D cultures of THP-1-derived foam cells and then detach these for seeding into 3D collagen hydrogels. Alternatively, untreated THP-1 cells could be seeded directly into 3D collagen hydrogels and then differentiated in situ to produce 3D foam cell cultures. Preliminary studies optimized the culture conditions to successfully generate THP-1-derived foam cells in 2D culture, and then explored whether these could be successfully retrieved via detachment using either a commercial Macrophage Detachment Solution (Promocell), accutase, EDTA/Trypsin, or cell scrapers. After treatment, the cell suspension was collected and the number and viability of foam cells was assessed using trypan blue and Oil Red O staining. These pilot studies found that none of the treatments identified above were successful in detaching significant numbers of THP-1-derived foam cells from the flask, with the majority being found remaining on the surface of the flask. Additionally, those cells that were successfully detached into the cell suspension were principally found to be non-viable. Therefore, efforts focused on seeding THP-1 cells into collagen hydrogels and differentiate these in situ into THP-1-derived foam cells.

The initial experiments assessed the potential of THP-1 cells to be differentiated into M1 macrophages within a type I collagen hydrogel as a precursor to foam cell formation. An M1 macrophage phenotype was selected as these cells are found to predominate in rupture-prone plaques [27]. PMA was added to the hydrogel solution to give a final concentration of 50 nM and cultured for 48 h. These collagen hydrogels were then subjected to live/dead cell staining, which found that cell viability remained high (in PMA-treated collagen hydrogels (Figure 2A)). To induce M1 differentiation of the M0 macrophages elicited by PMA treatment, the collagen hydrogels were rested by incubation in PMA-free media for 24 h, and then treated with fresh RPMI media containing 100 ng/mL LPS and 20 ng/mL IFN-γ as previously optimized for 2D foam cell culture. To assess differentiation of the THP-1 cells to M1 macrophages, immunofluorescent imaging was then performed following labelling with a primary antibody to CD80 and a FITC-labelled secondary antibody. To ensure that the CD80 labelling was specific for M1 macrophages, immunofluorescent imaging was compared in hydrogels exposed to LPS and IFN-γ (M1 macrophages) and those in which these polarizing cytokines were omitted (M0 macrophages). To ensure that the CD80 labelling was specific for M1 macrophages, experiments compared immunofluorescent labelling achieved in M1 and M0 macrophages cultured within the 3D hydrogels. As shown in Figure 2B, CM1 macrophages showed strong immunofluorescent labelling, indicating that M1 differentiation had successfully occurred. In contrast, weaker staining was observed in the hydrogels containing M0 macrophages (Figure 2B,C). These data, therefore, demonstrate that it was possible to generate M1 macrophages within the 3D collagen hydrogel.

### 3.2. Differentiation of THP-1-Derived Foam Cells in the 3D Collagen Hydrogel

Previous studies have found that macrophage cells will transform into foam cells after taking up oxLDL from the culture medium [28]. Sterile oxLDL samples were produced through copper sulphate oxidation and dialysis. Previous studies have demonstrated that oxidized LDL has different absorbance properties compared to the non-oxidized LDL [29]. As shown in Figure 3A, the oxidized samples were visibly cloudy compared to the non-oxidized aliquots taken from the same sample and had a notable difference in the absorbance spectra. This was further supported by assessment of the oxidation state of the LDL samples using an oxLDL ELISA kit. This showed a significant increase in the presence of oxLDL within the samples compared to the non-oxidized LDL—indicating that the copper sulphate treatment was effective and had increased the presence of oxLDL in the samples (Figure 3B). These data, therefore, indicate that copper sulphate-mediated oxidation of the samples was effective and could be used as a substrate to trigger foam cell formation in the THP-1-derived M1 macrophages.

Experiments assessed whether further treatment of the M1 macrophage-containing 3D cultures with 50 μg/mL oxLDL was able to elicit the formation of THP-1-derived foam cells within the collagen hydrogels. Following cytokine treatment, fresh RPMI media containing 50 μg/mL oxLDL were added to the cells for 72 h. After this, the presence of foam cells within the 3D culture was assessed using Nile Red, a fluorescent stain that labels intracellular neutral lipids and has been used to characterize lipid uptake in culture macrophages [30]. As shown in Figure 3C–E, intracellular, neutral lipid droplets can be observed in THP-1 cells in the focal plan, indicating that oxLDL treatment was able to successfully elicit THP-1-derived foam cell production within the collagen hydrogels to produce a prototype 3D neointimal biomimetic hydrogel.

Whilst the neointimal biomimetic hydrogel developed in a simple collagen hydrogel could be used to introduce an atherosclerotic plaque into the tissue-engineered arterial construct, the density of the THP-1-derived foam cells within these constructs was relatively low compared to that observed in a fatty streak or more advanced lesions. Additionally, the collagen construct was relatively weak and would be challenging to handle and affix within the tissue-engineered arterial construct. Plastic compression has been used to improve the mechanical properties of hydrogel through squeezing excess aqueous fluid out of the gel, significantly reducing the thickness of the hydrogels. This increases the density of the remaining collagen and cells and increases the stiffness of the construct [31]. Experiments assessed whether viable THP-1-derived foam cells could be successfully developed within a collagen hydrogel after plastic compression. To achieve this, the THP-1 cells were seeded into collagen hydrogels and set in a mold. Plastic compression was performed by placing a sterile weight on top of the hydrogel for a short period of time, with filter paper underneath to absorb any extracted fluid. The compressed hydrogels were then placed into RPMI media and then treated sequentially with PMA, LPS, IFN-γ, and oxLDL, as described previously, for the uncompressed gels. Live/dead cell staining of the compressed hydrogels found that the viability of the THP-1 cells was not significantly affected (89.3% ± 0.7%; Figure 4A). To assess foam cell production within the compressed collagen hydrogels, THP-1-derived M1 macrophages were prepared in compressed collagen hydrogels using the same culture method which was used for the non-compressed method. The samples were then treated with complete RPMI media containing either 50 μg/mL oxLDL or 50 μg/mL of non-oxidized LDL, and then incubated for a further 72 h. The gels were then fixed and stained with Nile Red stain using the same protocol which was used for the non-compressed hydrogels previously. As shown in Figure 4B intracellular neutral lipid droplets (red dots) could be observed in most of the cells in samples treated with oxLDL, but not in those treated with non-oxidized LDL. Nile Red fluorescence through the hydrogels was found to be significantly higher in the oxLDL-treated samples (348 ± 31 arbitrary units) than those treated with non-oxidized LDL (79 ± 9; *n* = 9; *p* < 0.05; Figure 4B).

### 3.3. Treatment with Atorvastatin Reduces the Accumulation of Lipids into the 3D Neointimal Biomimetic Hydrogel

M1 macrophages were characterized by the secretion of a variety of pro-inflammatory cytokines such as TNF-α and IL-6 [32]. The expression of these pro-inflammatory cytokines has previously been shown to be upregulated by treatment of THP-1-derived macrophages with oxLDL [33,34,35]. Experiments were conducted to assess whether the foam cells within the compressed collagen hydrogel were able to replicate these effects, by assessing cytokine secretion into conditioned media taken from hydrogels containing either THP-1-derived-foam cells or M1 macrophages. Conditioned media samples exposed for the same time frame to an acellular collagen hydrogel were used to control for basal presence of these cytokines in the FBS-supplemented media. Conditioned media samples from THP-1-derived foam cells contained significantly increased amounts of IL-6 and TNF-α compared to the acellular collagen hydrogel (Figure 5A,B)—consistent with the successful differentiation of THP-1 cells to proinflammatory macrophages. Conditioned media from hydrogels containing M1 macrophages were found to contain IL-6 and TNF-α, but only IL-6 was found to be significantly greater than that observed from the conditioned media from the oxLDL-treated hydrogels. The oxLDL loading of the THP-1-derived foam cells was seen to enhance the secretion of these pro-inflammatory cytokines compared to M1 macrophages, consistent with previous research, but this was only found to be significantly higher for IL-6.

### 3.4. Treatment with Atorvastatin Reduces the Accumulation of Lipids but Not the Proinflammatory Properties of the 3D Neointimal Biomimetic Hydrogel

Atorvastatin is widely used in the primary and secondary prevention of acute cardiovascular events in patients at risk of suffering thrombotic events. Previous studies have demonstrated that statins are able to inhibit the uptake of oxidized LDLs by foam cells. This potentially occurs due to a number of differing effects, including the regulation of foam cell scavenging receptor expression, a tendency for lipids with lower oxidation potential to be loaded into LDLs, and the direct anti-oxidant properties of statins reducing the oxidative state of the LDLs [36]. An Oil Red O staining assay was performed to assess whether atorvastatin treatment inhibited oxLDL uptake in the neointimal biomimetic hydrogel (Figure 5C). This study indicated that there is more lipid stored within the foam cells generated in methanol-treated conditions than those neointimal cultures treated with atorvastatin. These data, therefore, are consistent with previous observations that statins block oxLDL uptake into macrophages and foam cells. An ELISA assay was performed to assess whether preventing lipid loading into the foam cells had an impact on the pro-inflammatory properties of the 3D neointimal biomimetic hydrogel. No significant difference in IL-6 secretion could be observed in an assessment of IL-6 secretion into the conditioned media extracted from atorvastatin- and methanol-treated 3D neointimal cultures containing THP-1-derived foam cells (Figure 5D), indicating that this treatment did not reduce its secretion of this proinflammatory cytokine.

### 3.5. The 3D Neointimal Cell Culture Model Containing THP-1-Derived Foam Cells Is Able to Trigger Coagulation of Human Platelet Poor Plasma

Ruptured atherosclerotic plaques can trigger activation of the blood coagulation cascade through the presence of tissue factor present within the neointimal layer [37]. Tissue factor has been shown to be localized within foam cells present within the plaque [38]. Additionally, previous studies in 2D cultures of THP-1 cells have demonstrated that tissue factor activity of these cultures can be upregulated by treatment with either lipopolysaccharide [39] or oxidized low-density lipoprotein [40]. However, no previous studies have been conducted in THP-1 cells cultured within a 3D cell culture model; therefore, it is uncertain whether the 3D neointimal model can trigger blood coagulation and thus be used to study the processes of atherothrombosis. Previously, Musa et al. [22] developed a testing platform to assess the ability of tissue-engineered arterial constructs to activate human platelets. This was achieved through using a custom-made sodium acetate frame to hold the luminal surface of the construct on top of a washed human platelet suspension (Figure 1). This method was adapted to assess whether the 3D neointimal construct could initiate the activation of the blood coagulation cascade by performing a prothrombin time measurement on human platelet-poor plasma samples exposed to the luminal surface of neointimal constructs containing either THP-1-derived foam cells or M1 cells. As previous experiments have demonstrated an ability for fibrillar collagen to activate Factor XII [41], it is possible that the compressed collagen could trigger activation directly. Therefore, the cellular constructs were also compared against an acellular compressed collagen hydrogel as a control. As shown in Figure 6A, the foam cell-containing construct had the shortest prothrombin time (88 ± 8 s). This was significantly faster than both the M1 cell-containing gel (158 ± 13 s; *n* = 6; *p* < 0.05) and the acellular collagen hydrogel (377 ± 28 s; *n* = 6; *p* < 0.05). These data indicate that foam cells can trigger the blood coagulation system, consistent with previous studies in 2D cultures that showed that the atherogenic transformation of foam cells increases tissue factor expression [40]. Additionally, the M1 cells were found to trigger coagulation significantly faster than the acellular collagen hydrogel (*p* < 0.05), indicating that the LPS-stimulated M1 cells can also trigger blood coagulation.

Plastic compression increases the stiffness and the collagen concentration of the collagen hydrogels [42]. Previous studies have demonstrated that THP-1 cells take on a more pro-inflammatory phenotype when cultured upon stiffer collagen gels [43], suggesting that compression could enhance the pro-coagulant expression of the THP-1-derived foam cells. To assess this possibility, experiments were performed to compare the procoagulant properties of compressed or non-compressed collagen hydrogels. These were compared against a control in which a platelet-poor plasma (PPP) sample alone was recalcified. Compression of the collagen hydrogel did not significantly alter the prothrombin time (116 ± 12 s vs. 90 ± 5 s for compressed and non-compressed gels, respectively; both *n* = 5; *p* > 0.05; Figure 6B). These are both significantly longer than the prothrombin time measured from the untreated Ca^2+^-treated PPP samples (735 ± 12 s; *n* = 3; both *p* < 0.05 compared to compressed and non-compressed hydrogels). These data indicate that the change in stiffness or collagen concentration caused by compression of the 3D neointimal biomimetic hydrogel does not alter its procoagulant properties.

### 3.6. The 3D Neointimal Culture Model Possesses Measurable Extrinsic Factor Activity That Is Enhanced by Incubation with oxLDL

It has been established that macrophages and foam cells within atherosclerotic lesions possess significant tissue factor activity that underlies its thrombotic potential [44]. The prothrombin time measurements of the M1- and foam cell-containing hydrogels demonstrate that they are both capable of triggering blood coagulation. To investigate whether this activity could be due to enhanced tissue factor in the foam cells in the 3D neointimal biomimetic hydrogel, we used a previously developed microplate assay of extrinsic clotting factor activity [24]. This assay involves incubating the constructs with a solution containing inactive Factor VII (the target of tissue factor), and a fluorescent indicator of activated factor VII (FVIIa) activity, SN-17a. When digested by FVIIa, SN-17a becomes more fluorescent, and so the rate of fluorescence increase can be used to assess the tissue factor activity contained within each construct in the absence of any endogenously produced FVIIa. Previous experiments on samples containing human coronary artery smooth muscle cells have demonstrated that this fluorescence increase is dependent upon both the presence of a tissue factor-bearing construct and the presence of inactive Factor VII, demonstrating that the fluorescence increases are due to the presence of tissue factor in the sample [24]. However, previous studies have demonstrated that macrophages in atherosclerotic plaques are also able to produce FVII [45], indicating that fluorescent increases will be sensitive to the total activity of the extrinsic clotting factors, tissue factor, and factor VIIa, within this sample (Figure 6C). When both the M1- and foam cell-containing hydrogels were incubated with Factor VII and SN-17a, they elicited a significantly enhanced fluorescence increase during the incubation period compared to the acellular collagen gel. The fluorescent increase was found to be significantly enhanced for hydrogels containing foam cells compared to those containing M1 macrophages (*n* = 12; *p* < 0.05; Figure 6D). These results are consistent with previous studies that have demonstrated that oxLDL treatment of THP-1 cells significantly enhances their tissue factor activity (Owens et al., 2012). It is likely that this underlies the faster prothrombin times of the foam cell-containing hydrogel.

Previous studies have demonstrated that atorvastatin and other statins can inhibit tissue factor expression in monocytes [40,46]. Experiments were thus undertaken to assess whether atorvastatin reduced the procoagulant properties of the foam cell-containing biomimetic hydrogels. Treatment with atorvastatin significantly increased the measured prothrombin time of the neointimal biomimetic hydrogels compared to methanol-treated control samples 562 ± 68 s vs. 155 ± 28 s for atorvastatin- and MeOH-treated constructs, respectively; *n* = 8; *p* < 0.05; Figure 6E. In comparison, control samples in which the platelet-poor plasma used on each day was treated with calcium alone took 1711 ± 77 s (*n* = 6), indicating that treatment of the 3D neointimal biomimetic hydrogel with atorvastatin reduces the procoagulant properties of foam cells [40] and this model could be used to assess the ability of other drugs to reduce the prothrombotic potential of atherosclerotic plaques.

To further investigate, the procoagulant properties of the atorvastatin-treated constructs were assessed for extrinsic clotting factor activity using a modified SN-17a assay. Following the end of the culture period, the culture media were aspirated and the atorvastatin and methanol-treated neointimal biomimetic hydrogels were incubated with an HBS solution containing 1 mM CaCl_2_ and 10 nM inactive FVII for 30 min at room temperature. Additionally, as a control, the same solution was added to an empty well to demonstrate that the hydrogel was required for the activation. This solution was then removed from the hydrogel and placed in a fresh well with SN-17a to test for FVIIa activity arising either from endogenous activity of the foam cells or from activation of the inactive precursor by tissue factor activity arising from the biomimetic hydrogel. Additionally, a methanol-treated neointimal biomimetic hydrogel was incubated with an HBS solution containing 1 mM CaCl_2_ but no inactive FVII (FVII-), to help assess the level of endogenous FVIIa activity present in the hydrogel. This experiment demonstrated that there was an endogenous production of FVIIa observed from the FVII- sample. However, there was greater activity still in the methanol-treated sample exposed to exogenous FVII, indicating that the assay could detect FVIIa produced through tissue factor activity in the sample (Figure 6F). Interestingly, this level is significantly reduced by atorvastatin treatment, indicating that the activity of endogenously produced tissue factor and/or FVIIa is reduced by this treatment, consistent with the prolonged prothrombin time observed in these samples and suggesting that atorvastatin control of clotting factor activity in the neointima could underlie its ability to reduce the risk of acute cardiovascular events.

### 3.7. The 3D Neointimal Model Can Trigger a Slow Platelet Activation and Aggregation

Ruptured atherosclerotic plaques expose the bloodstream to adhesive and soluble ligands that can trigger platelet aggregation [47], so it would be expected that an effective 3D neointimal biomimetic hydrogel should trigger human platelet activation and aggregation. Due to the complex cellular and biochemical environment present in plaque materials obtained from surgical sources, it has not been possible to identify whether foam cells themselves are responsible for providing the pro-aggregatory potential of the atherosclerotic plaque or whether this results from the smooth muscle cells within the fibrous cap or indirectly through the production of thrombin from the coagulation cascade. The ability to engineer the cellular composition of the neointimal biomimetic hydrogel and to isolate it from plasma-based clotting factors provides an opportunity to assess whether THP-derived M1- and foam cells contribute to the pro-aggregatory potential of the neointimal biomimetic hydrogel. This was assessed using a cuvette-based system to expose our 3D neointimal culture models to washed human platelet suspensions [22]. Washed human platelets were utilized to remove the potential for platelet activation caused by thrombin generation elicited by the activation of the coagulation cascade. Washed human platelet suspensions were preincubated in contact with the luminal surface of the construct for 10 min at 37 °C under constant magnetic stirring. A sample was then transferred from the cuvette into a Chronolog aggregometer for further monitoring by light transmission aggregometry (Figure 7A). The acellular collagen hydrogel was not observed to elicit any significant platelet aggregation in any of the experiments (0.6 ± 0.4%), consistent with previous observations [22]. In contrast, preincubation with either the M1- or foam cell-containing constructs was able to elicit platelet aggregation; however, it was found to be slow at the onset, with the time taken to transition from shape change into aggregatory responses occurring slowly after the onset of monitoring on the aggregometer. This transition generally occurred earlier for foam cell-containing gels (437 ± 157 s) compared to M1-cell-containing constructs (610 ± 320 s), although this difference was not significant (*n* = 7; *p* = 0.11; Figure 7A,B). However, the maximum aggregation percentage measured over the 15 min recording period was found to be greater in the foam cell-containing hydrogels (59.9 ± 10.3%), although this was also not statistically greater than that elicited by the M1-containing hydrogels (41.0 ± 14.6%; *n* = 7; *p* > 0.05; Figure 7C). These data therefore indicate that, whilst both foam cell- and M1 macrophage-containing gels can consistently elicit aggregation, these responses are generally slow and variable in their time of onset, suggesting that these gels are only able to weakly activate human platelets in the absence of thrombin generation.

As activated human platelets secrete ATP-containing dense granules into the extracellular medium, a luciferin–luciferase dense granule secretion assay was performed on samples of the washed human platelet suspensions exposed to the 3D neointimal culture models to further assess the degree of platelet activation. Foam cell-containing hydrogels elicited a significant increase in luciferase luminescence in washed human platelet suspensions compared to samples exposed to acellular collagen gels (*n* = 7; *p* < 0.05; Figure 7D). M1-containing gels also elicited an intermediate increase in luciferase light production compared to the foam cell-containing and acellular collagen hydrogels, respectively; however, this difference was insignificant compared to both other samples (*p* > 0.05 for both). These results are consistent with the aggregation data, with the greatest increase in luciferin luminescence elicited by foam cells containing gels correlating with the most rapid platelet aggregation.

Atorvastatin has been shown to reduce human platelet function [48,49]. Therefore, the aggregometry experiments were repeated to compare the effect of atorvastatin- and methanol-treated neointimal biomimetic hydrogels on the responses observed. Atorvastatin treatment generally inhibited the aggregatory response observed, although this varied between blocking aggregation completely (Figure 7E) to slowing the rate of onset of aggregation (Figure 7F). Treatment tended to reduce the maximum platelet aggregation observed (35.0 ± 2.0%) compared to the methanol control sample (72.0 ± 17.6%; both *n* = 6), although this effect was not statistically significant (*p* = 0.07; Figure 7G). However, despite the variability in the responses, atorvastatin treatment was found to significantly increase the latency of the onset of platelet aggregation (681 ± 140 s vs. 434 ± 129 s for atorvastatin- and MeOH-treated constructs, respectively; *p* < 0.05; Figure 7H)—indicating that atorvastatin treatment can also reduce the weak pro-aggregatory properties of the 3D neointimal biomimetic hydrogel.

## 4. Discussion

In this paper, we demonstrate that a 3D neointimal biomimetic hydrogel can act as an effective prothrombotic substrate for a humanized in vitro model of atherothrombosis. A similar culturing methodology to that utilized in this paper has been independently reported to produce atherosclerosis plaque models [50]. This group also cultured THP-1-derived foam cells within a non-compressed collagen hydrogel using PMA, LPS, IFN-γ, and oxLDL to trigger foam cell differentiation in this 3D culture model. The ability of two groups to successfully use the same culturing strategy demonstrates that this technique is reliable and reproducible between labs and provides an effective methodology for producing a humanized in vitro neointimal model. Although this group has previously reported a method to differentiate THP-1 cells into foam cells within uncompressed collagen hydrogels, this paper represents the first demonstration of producing THP-1-derived foam cells within a compressed collagen hydrogel. This provides a high-density collagen matrix like that found in established plaques, in which up to 60% of plaque protein has been found to come from collagen [51]. The collagen density has been shown to play a key role in determining plaque stability [52,53]. Additionally, as macrophage proinflammatory responses improve as stiffness of the extracellular matrix increases [54], it is likely that the stiffer compressed collagen hydrogel will provide an optimal environment for recreating the proinflammatory conditions that lead to plaque rupture. However, whilst a type I collagen gel provides an effective basis for creating a biomimetic model of the neointima, the native plaque has a more complex extracellular matrix environment in which other collagen isoforms, elastin, and proteoglycans play a key role in regulating disease progression [55]. Therefore, future work will be required to develop this simple model through either exogenous incorporation of these molecules in the hydrogel scaffold or through stimulating their endogenous production by cells cultured within the neointimal model [24].

Beyond replicating the molecular and cellular components found in atherosclerotic plaques, to be functional, the neointimal biomimetic layer must also replicate the atherothrombotic events that underlie myocardial infarction and stroke. In this paper, we have demonstrated that the 3D neointimal biomimetic hydrogel possesses strong procoagulant activity as well as weaker pro-aggregatory properties. These results are consistent with previous studies that have reported the presence of significant tissue factor activity in in vivo atherosclerotic plaques [38,56,57,58,59]. This tissue factor activity has been shown to be associated with foam cells and vascular smooth muscle cells, as well as microparticles created by apoptosis of foam cells within the plaque [57,60,61]. Wilcox et al. (1989) [38] have reported that the necrotic core, the principal location of foam cells within the atherosclerotic plaque, is a particularly strong source of TF. These data suggest that a valid neointimal biomimetic hydrogel should possess significant tissue factor activity. SN-17a experiments demonstrated that the 3D neointimal biomimetic hydrogel possesses measurable tissue factor and FVIIa activity that can trigger the rapid activation of the coagulation cascade in PPP. These provide the first experimental evidence that a tissue-engineered human atherosclerotic plaque can replicate the ability of an in vivo plaque to trigger the activation of the hemostatic system. Additionally, our data confirm previous findings according to which treatment with oxLDL can upregulate tissue factor activity of both THP-1-derived foam cells [62,63,64]. This has been shown to occur through oxLDL-mediated activation of the activity of the NF-κβ tissue factor [49,65]. By augmenting this signaling pathway, it may be possible to further enhance the expression of tissue factor activity in these constructs. For example, previous studies have demonstrated that tissue factor activity can be augmented in THP-1 cell cultures through inclusion of Monocyte chemoattractant peptide-1 [66]. Thus, further optimization of the culturing conditions may be able to further improve the thrombogenic properties of these constructs.

The neointimal biomimetic hydrogel can also induce slow platelet activation and aggregation. This effect is mediated by the presence of M1- and THP-1 cells within the constructs, as the acellular collagen hydrogel was not found to possess this same activity. This is consistent with previous studies that have demonstrated that the soluble collagen hydrogel does not possess the tertiary structure of collagen required to activate human platelets [22]. These results provide a first demonstration that foam cells can elicit platelet activation. This work showcases one of the advantages of this layer-by-layer fabrication method for creating our tissue-engineered arterial constructs as it allows us to investigate novel intercellular interactions such as this, by allowing us to selectively alter the composition of the neointimal layer and examine how this modifies the responses of blood cells and cells of the vascular wall. The mechanism by which the THP-1 activity is elicited is unclear—this could be due to the release of a soluble platelet activator, the synthesis of an adhesive ligand on the surface of the 3D neointimal construct, or directly providing a platelet-binding surface on the cell surface. The ability of M1 cells to inconsistently evoke this response indicates that this stimulus is likely to be present in M1 cells but is upregulated by oxLDL treatment. Due to the slow and variable speed of the activation of the response, it is unlikely that this is a traditional platelet activator that can trigger strong aggregatory responses, such as collagen or thrombin. Here, we considered whether the release of pro-inflammatory cytokines could act as a priming agent for platelet activation and aggregation. However, the ELISA data presented suggest that this is unlikely to be due to IL-6 as atorvastatin treatment does not impact on the release of this cytokine. However, other cytokines may be involved. TNF-α production has also been shown to be enhanced by oxLDL production in our cultures, and therefore this may represent another potential low-level activator of human platelets. Pignatelli and colleagues have documented a correlation between platelet activation and the plasma level of TNF- α among patients with heart failure in comparison with a control group [66]. This in vitro study reported a statistically significant decrease in the platelet activation after they treated the plasma of the study group with TNF-α antagonists. This indicates that the TNF-α could play a role in triggering platelet activation by these biomimetic hydrogels [66]. Alternatively, this may be related to an increase in adhesive ligands secreted into the hydrogels by foam cells. For example, previous studies have shown that macrophages can synthesize collagen VI [67], which can trigger human platelet activation [68]. However, it is unclear whether oxLDL application can upregulate collagen VI production in foam cells. Therefore, further experiments will be required to further assess whether soluble and/or adhesive ligands elicit platelet activation in response to exposure to the neointimal biomimetic hydrogels. Statins are known to decrease the morbidity and mortality of cardiovascular disease through a variety of different effects [69,70]. Here, we demonstrate that pre-treatment with atorvastatin significantly impacts the thrombogenic properties of the 3D neointimal constructs by reducing both the pro-aggregatory and procoagulant properties of the construct. This may be related to atorvastatin’s ability to reduce the accumulation of lipids within the neointimal biomimetic hydrogels. This work is consistent with previous studies that have reported that statin can decrease the intracellular lipid contents by stimulating the cholesterol efflux from foam cells [71,72]. As uptake of oxLDL has been shown to induce multiple changes in the cellular phenotype of foam cells [73], further assessment of the impact of atorvastatin on these genetic and epigenetic responses will be required to assess how atorvastatin has an impact on the thrombogenic properties of the biomimetic hydrogels.

When incorporated within our previously produced tissue-engineered arterial construct and 3D printed flow chamber, it will be possible to produce a thrombus-on-a-chip model that can be used to study and treat thrombotic events. This neointimal biomimetic hydrogel provides a different strategy to previous attempts to create a tissue-engineered model of human atherosclerosis, which have revolved around triggering monocyte recruitment and transepithelial migration into the neointimal space of a healthy blood vessel model [15,16,18]. In contrast, our strategy is to develop the neointimal environment outside of the blood vessel and then adapt our layer-by-layer fabrication strategy to introduce the tissue-engineered artery once this has been fully developed. This process allows us to artificially introduce a high density of foam cells directly into the model system without having to recruit and differentiate these from perfused blood samples. Of the previous attempts, only Zhang et al. (2020) have demonstrated that the recruited monocytes were able to differentiate into foam cells in the subendothelial space of their tissue-engineered artery [18]. However, our neointimal biomimetic hydrogel currently replicates the fatty streak, an early, clinically silent stage of atherosclerotic plaque development. Further development of the biomimetic hydrogel will be required to allow it to produce a construct with a fibrous cap, necrotic core, and calcification, which are characteristic features of advanced fibroatheroma stages of plaque development [52]. For example, the fibrous cap could be developed either exogenously seeding the intimal surface of the construct with HCASMCs or via the ability of the model to trigger endogenous vascular smooth muscle cell migration from the underlying medial layer when incorporated into the complete tissue-engineered arterial construct. Preliminary experiments co-culturing the neointima and medial layer together have indicated that the neointimal biomimetic hydrogel is able to trigger HCASMC migration toward the neointimal layer—although further work will be required to confirm these initial findings.

## 5. Conclusions

In this paper, we demonstrate the production of a 3D neointimal biomimetic hydrogel that possesses the prothrombotic properties of the native plaque. This represents the first demonstration of a tissue-engineered atherosclerosis model that can act as an effective thrombotic trigger. Interestingly, these pro-thrombotic effects were inhibited by pre-treatment of the 3D neointimal constructs with atorvastatin, consistent with the known ability of this statin to reduce the incidence of acute cardiovascular events in patients at risk of myocardial infarction. These studies therefore demonstrate the potential capability of the developed neointimal model as an in vitro humanized drug testing platform to assess the impact of novel drugs and drug combinations on atherothrombosis. When combined with our tissue-engineered arterial construct in the future, this will produce an experimental system that can be used to screen the effects of drugs and drug combinations on preventing atherothrombotic events prior to clinical trials, helping to reduce the use of animal models as well as the significant financial costs of failed clinical trials.

## Figures and Tables

**Figure 1 biomimetics-09-00372-f001:**
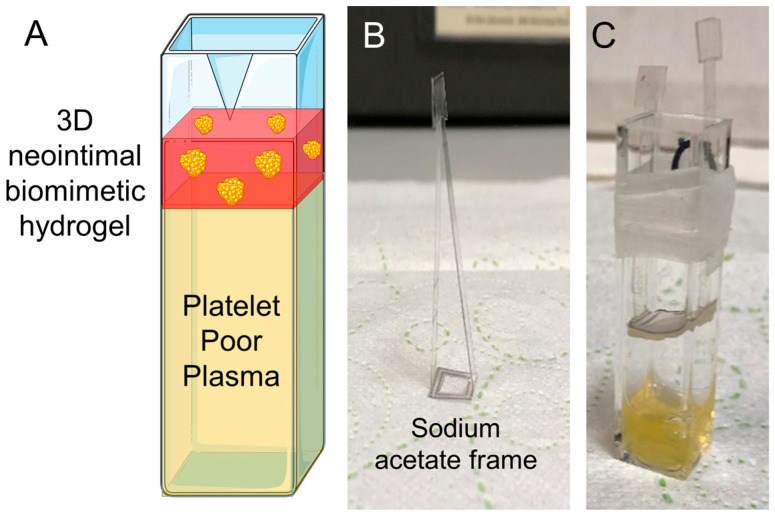
Experimental set-up for assessing prothrombin time stimulated by 3D neointimal culture models. (**A**) Diagram of the experimental set-up. Washed human platelets or platelet-poor plasma (PPP) were placed in contact with the luminal surface of the 3D neointimal culture model held in position using a sodium acetate frame. The cuvette was then held in a water bath at 37 °C under constant magnetic stirring. (**B**) Picture of the sodium acetate frame. (**C**) Photos of complete set-up with frame not fully inserted. Parafilm is placed around the outside of the cuvette to secure it in a cuvette holder in the water bath. Cuvette and foam cell icons obtained from Bioicons.com.

**Figure 2 biomimetics-09-00372-f002:**
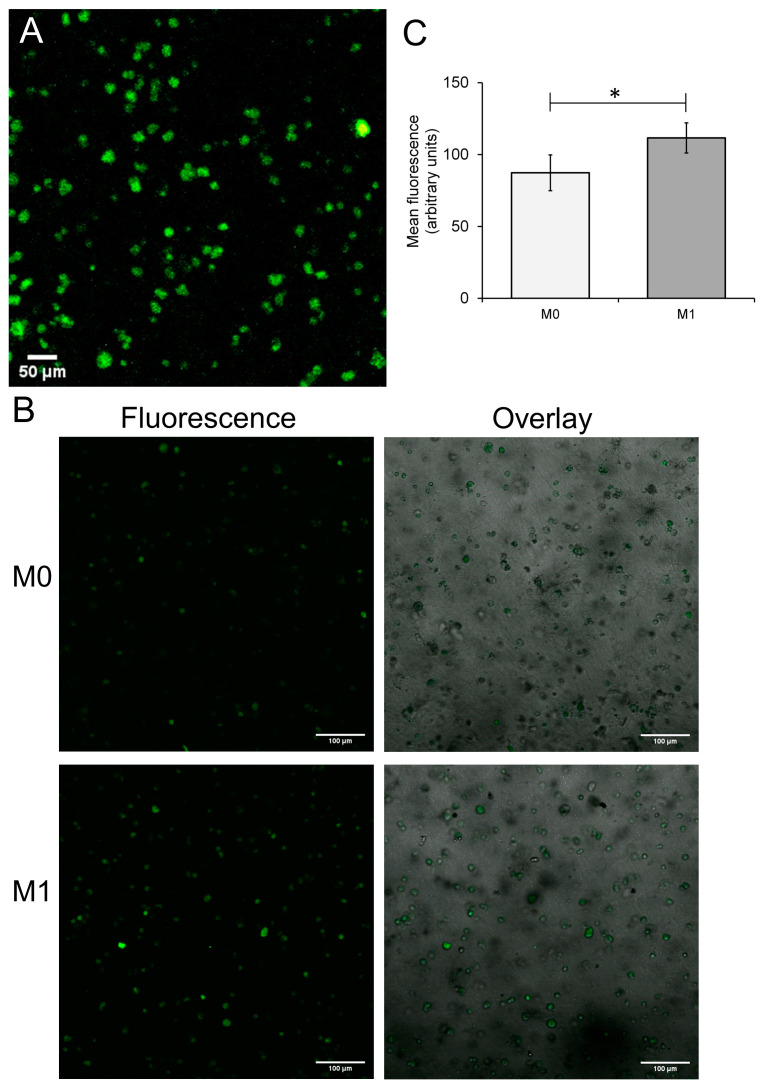
Culturing THP-1-derived M1 macrophages within a 3D collagen hydrogel. (**A**) THP-1 cells were mixed at a cell density of 3 × 10^6^ cells/mL into a neutralized solution of 3 mg/mL type I collagen. Following setting, the hydrogel was incubated with a supplemented RPMI medium containing 50 nM PMA and then incubated for 48 h at 37 °C, 5% CO_2_. Hydrogels were stained by a live/dead staining solution and imaged by confocal microscopy. (**B**) THP-1 cell suspensions were seeded at 6 *×* 10^5^ cells/hydrogel in RPMI media with 50 nM PMA. Cells were then incubated at 5% CO_2_, 37 °C for 48 h. Cells were rested by incubation in a PMA-free media for 24 h, and then cells were transferred into fresh RPMI containing either (**Top**) 100 ng/mL LPS and 20 ng/mL IFN-γ or (**Bottom**) their vehicle, DMSO for a further 72 h. Collagen hydrogels were then fixed and then immunolabelled using an unconjugated CD80 antibody and a FITC-labelled secondary antibody. Samples were then subjected to z-stack imaging using a confocal microscope. An x-y slice from the confocal slice with the highest mean fluorescence is shown along (**left**) and overlaid over the transmitted light image (**right**) (**C**) Image J analysis of the mean slice FITC fluorescence through the hydrogel. *n* = 6, * indicates *p* < 0.05.

**Figure 3 biomimetics-09-00372-f003:**
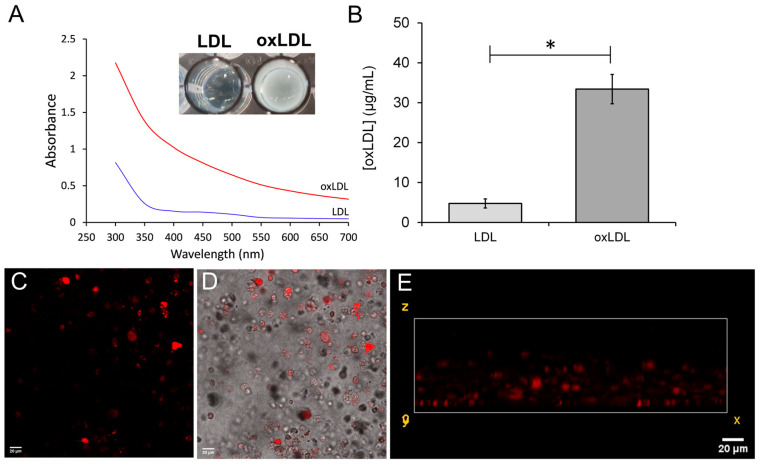
Culturing THP-1-derived foam cells in a 3D collagen hydrogel. (**A**,**B**) Oxidized and non-oxidized LDL samples were assessed to validate effective oxidation of the LDL particles. (**A**) Absorbance spectra of the oxidized (oxLDL) and untreated LDL (LDL) samples with inset of picture of the oxidized and non-oxidized samples in a 96-well plate. (**B**) Measurement of oxidative state using an oxLDL ELISA assay kit. * indicates *p* < 0.05 relative to the LDL control. *n* = 4. (**C**–**E**) THP-1 cell suspensions were seeded at density of 6 *×* 10^5^ cells/collagen hydrogel and cultured in RPMI media with 50 nM for 48 h. Cells were rested by incubation in a PMA-free media for 24 h, and then cells were treated with LPS and IFN-γ for 72 h. Lastly, collagen hydrogels were treated with 50 µg/mL oxLDL at 5% CO_2_, at 37 °C for 72 h. Collagen hydrogels were fixed with 10% formalin overnight and then were stained with Nile Red to assess the intracellular accumulation of neutral lipid through the sample by confocal microscopy. Representative images are shown from 4 experiments. (**C**) Fluorescent x-y slice through the hydrogel. (**D**) Fluorescent slice overlaid on the corresponding transmitted light image. (**E**) Fluorescent x-z slice from a z-stack analysis of the hydrogel.

**Figure 4 biomimetics-09-00372-f004:**
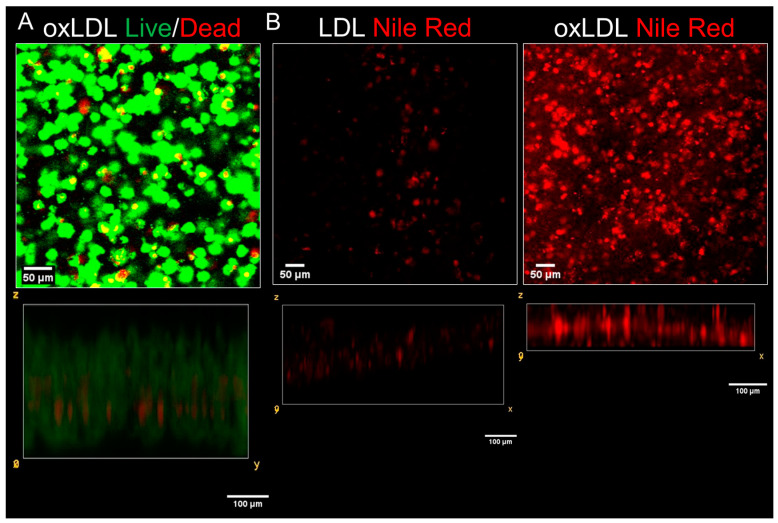
THP-1-derived foam cells could be produced within compressed collagen hydrogels that produce measurable concentrations of pro-inflammatory cytokines. THP-1 cell suspensions were seeded into compressed collagen hydrogels, and then incubated in RPMI media with 50 nM PMA at 5% CO_2_, 37 °C for 48 h. Cells were rested by incubation in a PMA-free media for 24 h, and then cells were treated with LPS and IFN-γ for 72 h. (**A**) The samples were then treated with 50 µg/mL oxLDL at 5% CO_2_, 37 °C for 72 h. After this period, cells were stained with Live/Dead Cell staining kit stain to assess cell viability with green indicating live cells, and red indicating dead cells. Images were taken using confocal microscopy. The **top** panel shows an x-y slice through the gel. The **bottom** panel shows an x-z slice through the hydrogel. (**B**) Following addition of LPS and IFN-γ, samples were then treated with either 50 µg/mL LDL (**Middle**) or 50 µg/mL oxLDL (**Right**) at 5% CO_2_, 37 °C for 72 h. After this period, cells were fixed and stained with Nile Red (Red) to assess intracellular accumulations of neutral lipids. The top panel shows an x-y slice through the gels, whilst the bottom panel shows an x-z slice through the gels.

**Figure 5 biomimetics-09-00372-f005:**
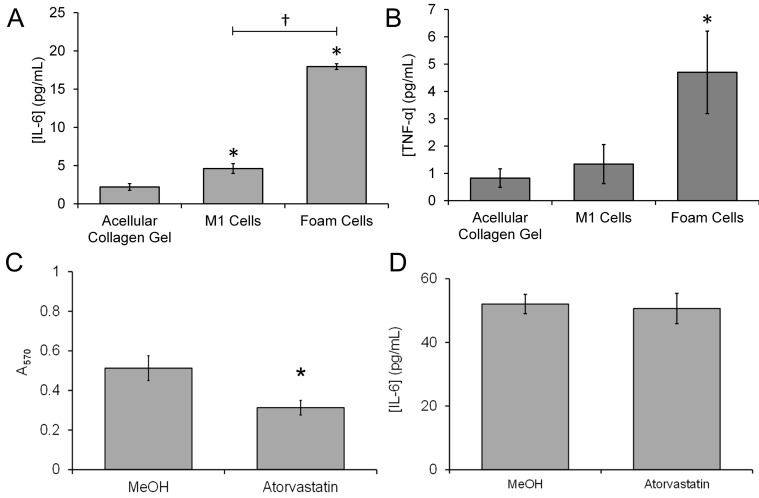
Atorvastatin treatment reduces lipid accumulation within THP-1-derived foam cells within the 3D neointimal co-culture model. Three-dimensional collagen hydrogels containing M1 cells or foam cells were produced. Acellular collagen gels were also placed in culture alongside these constructs as a control sample. Conditioned media samples were then extracted and subjected to ELISA analysis of either (**A**) TNF-α or (**B**) IL-6 contained within the samples. * = *p* < 0.05 relative to the acellular control. † indicates *p* < 0.05 compared to the indicated samples. (**C**) Three-dimensional neointimal culture models with THP-1-derived foam cells were either treated with methanol (MeOH) or 20 µM atorvastatin at the same time as oxLDL. At the end of the culture period, the gel was then removed and fixed with paraformaldehyde for an Oil Red O assay. The fixed gels labelled with Oil Red O and washed three times to remove dye that remained unbound to intracellular lipids. After washing the remaining dye bound to intracellular lipids, it was leached out of the sample by incubation with isopropanol. The concentration of Oil Red O dye in the isopropanol sample was then measured by loading into a 96-well plate and using a BioTek synergy 2 microplate reader to assess sample absorbance at 570 nm. (**D**) A sample of conditioned media was also taken from the atorvastatin-treated cultures for an IL-6 ELISA assay. *n* = 6, * indicates *p* < 0.05 relative to the methanol-treated control.

**Figure 6 biomimetics-09-00372-f006:**
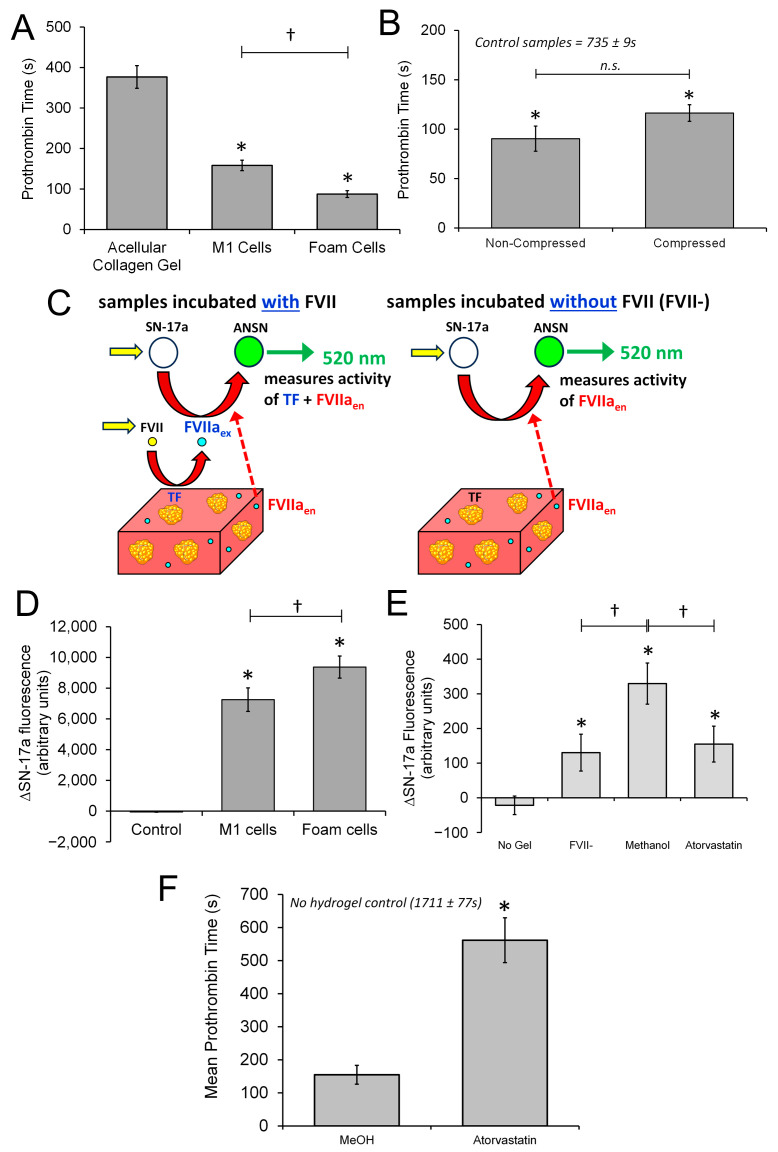
Treatment with atorvastatin reduces the procoagulant properties of the THP-1-derived foam cell constructs. (**A**) Three-dimensional neointimal culture models were produced in which THP-1-derived foam cells or M1 cells were cultured, as well as a cell-free collagen hydrogel. (**B**) THP-1-derived foam cells were cultured within either a compressed or a non-compressed collagen hydrogel. Samples were then placed onto sodium acetate frames and placed into contact with the surface of prewarmed, recalcified human platelet-poor plasma (PPP). The sample was kept warm in a 37 °C water bath. The time taken for fibrin formation after addition of the 3D neointimal cultures models was recorded for each sample. Values shown are Mean ± SEM of 5 independent experiments. * indicates *p* < 0.05 vs. the control sample. n.s. indicates not statistically significant. (**C**) Tissue factor assay methodology. A solution of inactive factor VII, CaCl_2_, and the fluorescent Factor VIIa indicator, SN-17a, were added onto foam cell and M1-containg neointimal culture models, or an acellular collagen set within 48-well plates immediately before the start of the recording. Tissue factor in the sample activates Factor VII to Factor VIIa, which in turn can cleave SN-17a, releasing the fluorescent moiety ANSN moiety which can be measured by the microplate reader. (**D**) Culture media were removed and an HBS solution containing 25 nM inactive Factor VII, 1 mM CaCl_2_, and 2.5 µM SN-17a was added to acellular collagen gels, or M1- or THP-1-containing 3D neointimal culture models. The well plate was then read fluorometrically for 10 min. The mean change in SN-17a fluorescence was observed between the initial reading and that observed after 10 min for TEML. Values shown are Mean ± SEM of 12 independent experiments. * indicates *p* < 0.05 vs. the control sample. † indicates *p* < 0.05 relative to the indicated sample. (**E**,**F**) Three-dimensional neointimal culture models were produced with THP-1-derived foam cells which were either treated with methanol (MeOH) or 20 µM atorvastatin at the same time as the oxLDL. (**E**) After the culturing period, samples were placed onto sodium acetate frames and placed into contact with the surface of prewarmed, recalcified human platelet-poor plasma (PPP). A negative control was also conducted in which acellular collagen hydrogels were also monitored for procoagulant activity. The sample is kept warm in a 37 °C water bath. The time taken for fibrin formation after addition of the 3D neointimal cultures models was recorded for each sample. Values shown are Mean ± SEM of 3 experiments. * indicates *p* < 0.05 vs. the methanol-treated sample. (**F**) At the end of the culture period, conditioned media were collected and mixed with a CaCl_2_-containing HBS and inactive Factor VII. Methanol-treated samples were also treated with Ca^2+^-containing HBS without inactive FVII (FVII-). Samples were incubated for 30 min at room temperature. Supernatant was then moved to another well containing SN-17a and fluorescence increases were then recorded for 15 min in a BioTek Synergy 2 microplate reader. A bar chart summarizing the mean increase in SN-17a fluorescence observed in methanol- and atorvastatin-treated constructs. * indicates *p* < 0.05 when compared to No gel control samples. † indicates *p* < 0.05 between indicated samples.

**Figure 7 biomimetics-09-00372-f007:**
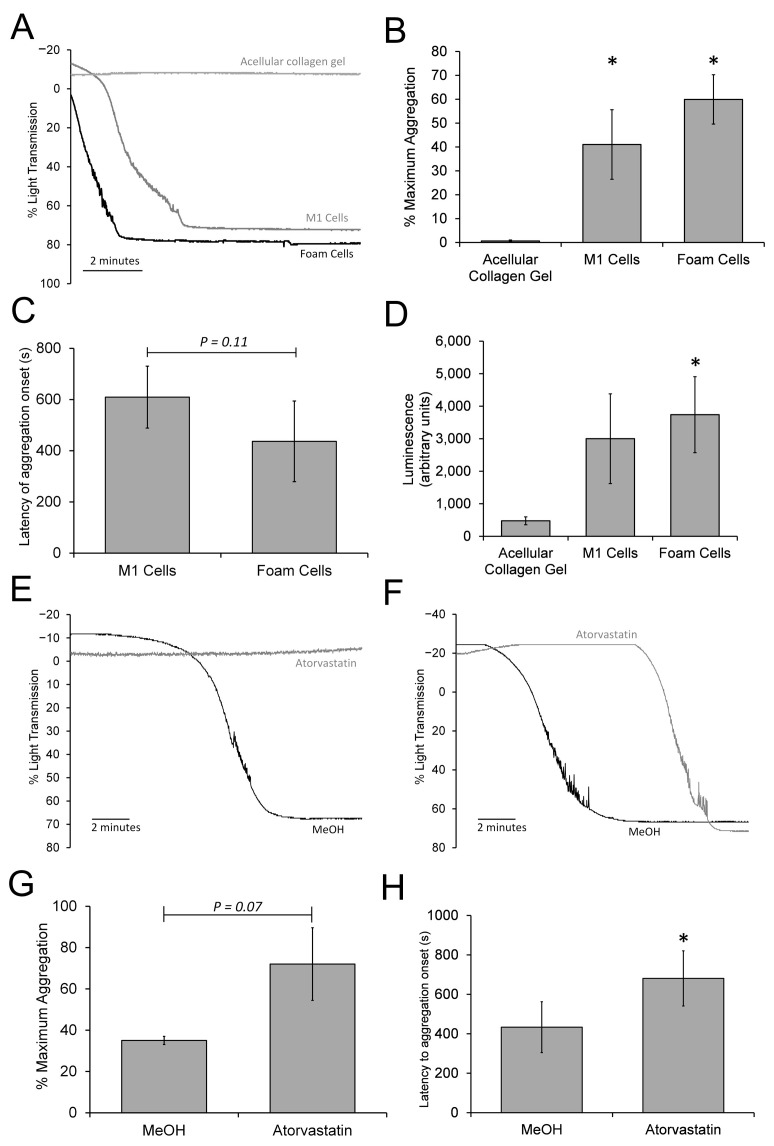
Atorvastatin reduces the slow platelet activation and aggregation triggered by the neointimal biomimetic hydrogel. (**A**) Washed human platelet suspensions resuspended in supplemented HBS containing 1 mM CaCl_2_ were exposed to either an acellular compressed collagen hydrogel, or a 3D neointimal culture model containing either THP1-derived M1 or foam cells. The constructs were incubated for 10 min at 37 °C under constant magnetic stirring. The gel was then removed and the samples were transferred for monitoring on a light transmission aggregometer at 37 °C under constant magnetic stirring for up to 16 min. A representative aggregometry trace is shown. Bar charts summarizing the mean maximum aggregation percentage observed during the recording period (**B**) and the mean time taken for the samples to begin to aggregate. (**C**) *n* = 7, * indicates *p* < 0.05 relative to the acellular collagen hydrogel. (**D**) The remaining platelet solution was transferred back to the water batch and incubated for a further 10 min. After this, a sample was pipetted into a 96-well platelet and mixed with 20% [*v*/*v*] luciferin–luciferase, and luminescence read for 1 min on a BioTek synergy 2 microplate reader. *n* = 7, * indicates *p* < 0.05 relative to the acellular collagen hydrogel. (**E**,**F**) Washed human platelet suspensions resuspended in supplemented HBS containing 1 mM CaCl_2_ were exposed to either an atorvastatin- or methanol (MeOH)-treated neointimal construct containing foam cells. The constructs were incubated for 10 min at 37 °C under constant magnetic stirring and were monitored on a light transmission aggregometer as described above for up to 16 min. This drug’s effect on the aggregatory response was found to be variable between experiments and donors. (**E**) Experiment in which atorvastatin treatment completely blocked platelet aggregation, whilst in (**F**) the response was slowed. Bar charts summarizing the mean maximum aggregation percentage observed during the recording period (**G**) and the mean time taken for the samples to begin to aggregate (**H**). * indicates *p* < 0.05 relative to the methanol-treated control sample.

## Data Availability

The data presented in this study are available in the article.

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
