# Peer review of "Developing a Biomimetic 3D Neointimal Layer as a Prothrombotic Substrate for a Humanized In Vitro Model of Atherothrombosis"

_biomimetics, 2024, doi:10.3390/biomimetics9060372_

Round 1

Reviewer 1 Report

Comments and Suggestions for Authors

I recommend acceptance of this paper as is.

Author Response

We thank the reviewer for their recommendation of our paper

Reviewer 2 Report

Comments and Suggestions for Authors

The manuscript is well organized, but I suggest to add more data abouth thrombotic problem.

More information regarding the research background will be interesting for readers.

Author Response

We thank the reviewer for their positive and constructive comments.  Due to the short timeline for revision of the manuscript it has not been possible to add additional data about the thrombotic problems.

In line with another reviewer’s comments, we have added additional discussion around the extracellular matrix composition of the native atherosclerotic plaque on L763-776 of the manuscript, which we hope will provide additional useful research background to the project for readers.

Reviewer 3 Report

Comments and Suggestions for Authors

In the manuscript titled “Developing a biomimetic 3D neointimal layer as a prothrombotic substrate for a humanised in vitro model of atherothrombosis”, Jassim et al produced a biomimetic hydrogel of the neointima by culturing THP-1-derived foam cells within 3D collagen hydrogels. The research design is appropriate and the results are clearly presented at the most of the time. However, some results need to be improved. For example, in Figure 2, the staining of THP-Q1 cells in both B and C is very weak, I can't tell any differences. If z-stack imaging failed, the authors can also show xy-axis image. While the design of this biomimetic system seems to be too simple, at least the author should address out the importance of using collagen but not other polymer as their hydrogel network (also with sufficient data to prove). One more advice, the label of each image in the figure (A, B, C……) should only appear ONCE in the figure legend for better readability.

Author Response

Reviewer 3: In the manuscript titled “Developing a biomimetic 3D neointimal layer as a prothrombotic substrate for a humanised in vitro model of atherothrombosis”, Jassim et al produced a biomimetic hydrogel of the neointima by culturing THP-1-derived foam cells within 3D collagen hydrogels. The research design is appropriate and the results are clearly presented at the most of the time. However, some results need to be improved.

For example, in Figure 2, the staining of THP-Q1 cells in both B and C is very weak, I can't tell any differences. If z-stack imaging failed, the authors can also show xy-axis image.

We thank the reviewer for highlighting the unclear data presentation. We have utilized the referees suggestion and switched the  z stack images, with the xy slice with the highest mean fluorescence reading from M0 and M1 hydrogels from the same experimental repeat. We have updated Figure 2 and its figure legend accordingly.

While the design of this biomimetic system seems to be too simple, at least the author should address out the importance of using collagen but not other polymer as their hydrogel network (also with sufficient data to prove).

We thank the reviewers for their comments. As collagen provides a significant proportion of the protein content of plaques, the use of a type I collagen hydrogel provides a basic replica of the bulk of the native neointima and provides an accessible and simple system in which other investigators can replica and develop this initial study.   However, we agree with the reviewer that solely using type I collagen only provides a simplified version of the extracellular matrix found in atherosclerotic plaques with recent data suggesting key roles for elastin and proteoglycans in determining plaque development.  As suggested by the reviewer, we have added text to state this as a current limitation of the study and to highlight other key previous data on this topic in the discussion between L763-776.

One more advice, the label of each image in the figure (A, B, C……) should only appear ONCE in the figure legend for better readability.

We have amended the figure legends throughout the paper to ensure that there is one specific reference to each panel in the figure legends.

Reviewer 4 Report

Comments and Suggestions for Authors

In this work, the authors thoroughly present the development of a biomaterial system that could be implemented as an in vitro model of atherothrombosis. The primary focus of the study was to use collagen type I for the creation of a 3D neointimal layer, which could mimic an early-stage atherosclerotic plaque with pro-thrombotic properties. The rationale for each experiment was clearly laid out, as well as the materials and methods employed. Exhaustive characterization was conducted to validate the ability of the hydrogel to provide a platform for the production of THP-1 derived foam cells, as well as a procoagulant substrate (platelet aggregation and activation). The functionality of this model was further evidenced upon testing with atorvastatin.   

Minor comments:

1.     There are syntax errors throughout the document. Please, revise. For example (but not limited to):

-          Lines 33-34: …system generates occlusive thrombi that blocks the downstream coronary or cerebral arteries that lead to heart attacks and strokes, respectively…

-          Line 78: …slides has prevented this technique from being widely adopted…

-          Lines 97-100: please, revise writing, as the last 2 lines do not convey a clear idea (..for tissue engineering advanced stages atherosclerotic lesions..)

-          Line 198: After culture, the collagen hydrogels were transferred to 24-well plates…

-          Line 200: Cells viability was assessed….

-          Line 260: ...(HBS) to a final cell density of to 2x108 cells/mL...

-          Line 430.

-          Lines 602-603.

2.     Also, there are some typos to be corrected (for example, check lines 351, 435, 437, 472, and 697). Please, revise.

3.     Section 2.3.4: Is there a specific reason why metallic and plastic molds were used? Did each kind of mold serve a different purpose? Please, clarify this in the revised version of the manuscript.

4.     Figure 2: what was the sample number used for image analysis of each group?

5.     Was the change in dimensions measured for the compressed hydrogels, relative to the uncompressed samples? Was it significant? If available, please include this information.  

Comments on the Quality of English Language

1.     There are syntax errors throughout the document. Please, revise. For example (but not limited to):

-          Lines 33-34: …system generates occlusive thrombi that blocks the downstream coronary or cerebral arteries that lead to heart attacks and strokes, respectively…

-          Line 78: …slides has prevented this technique from being widely adopted…

-          Lines 97-100: please, revise writing, as the last 2 lines do not convey a clear idea (..for tissue engineering advanced stages atherosclerotic lesions..)

-          Line 198: After culture, the collagen hydrogels were transferred to 24-well plates…

-          Line 200: Cells viability was assessed….

-          Line 260: ...(HBS) to a final cell density of to 2x108 cells/mL...

-          Line 430.

-          Lines 602-603.

2.     Also, there are some typos to be corrected (for example, check lines 351, 435, 437, 472, and 697). Please, revise.

Author Response

Reviewer 4: In this work, the authors thoroughly present the development of a biomaterial system that could be implemented as an in vitro model of atherothrombosis. The primary focus of the study was to use collagen type I for the creation of a 3D neointimal layer, which could mimic an early-stage atherosclerotic plaque with pro-thrombotic properties. The rationale for each experiment was clearly laid out, as well as the materials and methods employed. Exhaustive characterization was conducted to validate the ability of the hydrogel to provide a platform for the production of THP-1 derived foam cells, as well as a procoagulant substrate (platelet aggregation and activation). The functionality of this model was further evidenced upon testing with atorvastatin.  

Minor comments:

  1. There are syntax errors throughout the document. Please, revise. For example (but not limited to):

-          Lines 33-34: …system generates occlusive thrombi that blocks the downstream coronary or cerebral arteries that lead to heart attacks and strokes, respectively…

-          Line 78: …slides has prevented this technique FROM being widely adopted…

-          Lines 97-100: please, revise writing, as the last 2 lines do not convey a clear idea (..for tissue engineering advanced stages atherosclerotic lesions..)

-          Line 198: After culture, the collagen hydrogels were transferred to 24-well plates…

-          Line 200: Cells viability was assessed….

-          Line 260: ...(HBS) to a final cell density of to 2x108 cells/mL...

-          Line 430.

-          Lines 602-603.

We would like to thank the reviewer for identifying these syntax errors.  We have now corrected these in the revised manuscript as shown in red. We have also performed a thorough proofread and corrected a number of other grammatical errors. 

  1. Also, there are some typos to be corrected (for example, check lines 351, 435, 437, 472, and 697). Please, revise.

We would ike to thank the reviewer again for highlighting these errors.  We have now amended the manuscript to improve the quality of the written English.

  1. Section 2.3.4: Is there a specific reason why metallic and plastic molds were used? Did each kind of mold serve a different purpose? Please, clarify this in the revised version of the manuscript.

Both the plastic and metal moulds used in the study are made to identical specifications and do not provide any specific purposes for the creation of the 3D neointimal model.  We have clarified this in the paper by adding a comment to this effect on L183 of the revised manuscript.

  1. Figure 2: what was the sample number used for image analysis of each group?

We thank the reviewer for spotting this omission.  We have updated the figure legend for Figure 2C to indicate that this image analysis was conducted across 6 samples of each type of hydrogel.

  1. Was the change in dimensions measured for the compressed hydrogels, relative to the uncompressed samples? Was it significant? If available, please include this information.

We have not formally measured the thickness of either the uncompressed or compressed neointimal hydrogels. We can confirm that compression did significantly reduce the thickness of the construct and have added a note to this regard on L430. Our aim was to assess whether it was feasible to use plastic compression of the hydrogels to increase the collagen concentration, cell density and stiffness of the hydrogel without impacting on foam cell production. As we subsequently assess the biological response of blood and blood products to contact with the luminal surface of the compressed hydrogel, the thickness of the gel is unlikely to significantly impact upon the results of our study.

Uncompressed hydrogels were produced by loading 200 μL of cell-containing neutral collagen solution onto a 1 cm2 filter paper frame – so we would expect these constructs to have a mean thickness of around 2 mm, although due to curvature of the surface of the droplet this would not be consistent across the construct.

Compressed gels were made by setting 2 mL neutral collagen solution containing THP-1 cells in moulds that were 2 cm deep and then subject to compression. We have previously performed some pilot studies using optical coherence tomograph to examine our tissue engineered medial layer. This was made in the same way as the neointimal model except for the addition of THP-1 cells instead of Human coronary artery smooth muscle cells [24]. These images indicated that the compressed hydrogel was approximately 200 μm in thickness after compression, which should give an indication of the degree of volume reduction achieved.  As we have not formally measured this for the neointimal model, we are not able to add a specific value into the manuscript at this time.

Comments on the Quality of English Language

We have addressed the issues raised here in comments 1 and 2 above